# Validation and Screening Capacity of the European Portuguese Version of the SUNFRAIL Tool for Community-Dwelling Older Adults

**DOI:** 10.3390/ijerph18041394

**Published:** 2021-02-03

**Authors:** Ana Filipa Cardoso, Elzbieta Bobrowicz-Campos, Luísa Teixeira-Santos, Daniela Cardoso, Filipa Couto, João Apóstolo

**Affiliations:** 1Health Sciences Research Unit, Nursing, Nursing School of Coimbra, Portugal Centre for Evidence-Based Practice, A Joanna Briggs Institute Centre of Excellence, 3004-011 Coimbra, Portugal; dcardoso@esenfc.pt (D.C.); apostolo@esenfc.pt (J.A.); 2Centre of 20th Century Interdisciplinary Studies, Faculty of Psychology and Educational Sciences, University of Coimbra, 3000-115 Coimbra, Portugal; elzbieta.campos@gmail.com; 3Health Sciences Research Unit, Nursing, Nursing School of Coimbra, 3004-011 Coimbra, Portugal; luisasants@esenfc.pt; 4Alfena Hospital—Trofa Health Group, Health Sciences Research Unit, Nursing, Nursing School of Coimbra, 3000-232 Coimbra, Portugal; filipadccouto@gmail.com

**Keywords:** SUNFRAIL, psychometric properties, screening tool, frailty, older adults

## Abstract

Early detection of frailty may prevent or delay adverse health outcomes in community-dwelling older adults. In Portugal, there are currently no valid multidimensional frailty screening tools. SUNFRAIL is a user-friendly multidimensional tool for frailty screening that can be used in primary care. Aims: (i) to determine the validity and reliability of the European Portuguese version of the SUNFRAIL tool for use in community-dwelling older adults; (ii) to assess the screening capacity of this version of SUNFRAIL using Fried’s phenotypic model criteria for frailty as a reference test. Methods: Cross-sectional pilot study in a convenience sample of 128 community-dwelling older adults. Objective and subjective data were collected. Internal consistency, concurrent validity, sensitivity, and specificity (ROC curve analysis) were examined. Results: Internal consistency was low. Significant moderate to strong correlations were found between different domains and the total score. The differences between robust, pre-frail, and frail older adults were significant. SUNFRAIL was also correlated with multimorbidity. Sensitivity and specificity were satisfactory. Conclusions: The European Portuguese version of the SUNFRAIL tool is a promising frailty screening tool for community-dwelling older adults to be routinely used in clinical practice. However, more consistent results on its validity and reliability are needed to be used nationwide.

## 1. Introduction

The current COVID-19 pandemic has created new challenges for active and healthy aging [1]. Community-dwelling older adults are now more vulnerable and exposed to negative outcomes, and they have been forced to change their active and healthy aging habits. Recent studies [2,3,4] have shown that the measures applied to contain the coronavirus spread resulted in a relevant decrease in older adults’ physical activity, which negatively impacted their subjective well-being.

Geriatric care clinical settings also face a marked increase in demands for effective treatment of age-related clinical conditions, striving to provide personalized and timely comprehensive care. These demands are challenging even in normal times due to the high rates of multimorbidity in advanced age [5,6,7], which have become much more pronounced over the past year because the health systems’ resources had to be carefully distributed in line with pandemic-related priorities. Therefore, there is an urgent need to implement mechanisms that facilitate the shift from a disease-oriented to a preventive approach to ensure that people live independently and with quality of life for as long as possible and, consequently, contribute to health systems’ sustainability. One of the requirements for this shift is the use of tools for early diagnosis and treatment of age-related conditions, especially frailty.

As a common age-related condition, frailty is characterized by an increased vulnerability to adverse health outcomes that affect several domains of human functioning (physical, psychological, and social) [8], resulting from the decline in multiple physiological systems [9]. Due to its malleable nature, the condition of frailty can be reversed to a healthy state if the intervention focuses on the symptoms, adapting the treatment procedures to their clinical relevance and impact on functioning [10]. However, most of the available frailty screening tools do not address all domains of functioning, compromising frailty assessment and management. Early identification of frailty is essential for developing timely and tailored interventions based on evidence-informed clinical decision-making, but it requires easy-to-use instruments [11].

The multidimensional SUNFRAIL tool has a bio-psycho-social approach to address the gaps mentioned above [12]. It is a quick and easy-to-use tool, which facilitates its regular use in clinical practice. This nine-question tool assesses the presence of difficulties or problems in biological (items 1–5), psychological (items 6 and 7), and social domains (items 8 and 9). However, a recent study proposed a different categorization of this tool’s items [13]. SUNFRAIL items are scored 1 for “yes” and 0 for “no”, except for items 4 and 8, which are scored inversely. Higher scores suggest more frailty [12]. The SUNFRAIL tool proved to be a valid instrument for screening frailty in community-dwelling older adults [8]. Given these characteristics, the authors of this study proceeded with forward-backward translation and cross-cultural adaptation of the SUNFRAIL tool for European Portuguese, as recommended by international guidelines [14]. The detailed results of this process are published elsewhere [15].

The present study aimed (i) to determine whether this European Portuguese version of the SUNFRAIL tool is a valid and reliable instrument to be used in community-dwelling older adults, and (ii) to assess the screening ability of this SUNFRAIL version using Fried’s phenotypic model criteria for frailty as a reference test.

## 2. Materials and Methods

### 2.1. Participants

This cross-sectional study was conducted in a convenience sample of 128 community-dwelling older adults recruited by family nurses in cultural and sports associations, municipal services, and health and day centers in Portugal’s central region. The exclusion criteria were the presence of moderate to severe cognitive decline and unstable clinical condition. Data were collected from November 2018 to September 2019.

### 2.2. Instruments and Procedures

Sociodemographic and clinical characteristics: All eligible participants were asked to provide information on their sociodemographic characteristics (age, gender, marital status, and education level), anthropometric characteristics (weight and height), chronic conditions (neoplasms, blood and immune system disorders, endocrine and metabolic diseases, central nervous system diseases, special senses disorders, cardiovascular diseases, respiratory diseases, digestive diseases, skin diseases, musculoskeletal and connective tissue disorders, genitourinary disorders, hyperthyroidism, hypothyroidism, hypertension, restless legs syndrome, narcolepsy, obstructive sleep apnea, mental and behavioral disorders, or anxiety), and medication intake (anxiolytics, antihypertensives, beta-blockers, hypnotics, corticosteroids, anti-inflammatory drugs, melatonin receptor agonists, thyroid hormones, muscle relaxants, antipyretics, or melatonin). They were also asked to perform tasks to identify symptoms of physical frailty and assess their cognitive status.

Measures of frailty: All participants were screened for frailty based on Fried’s frailty phenotype criteria [16] and completed the European Portuguese version of the SUNFRAIL [15]. Fried’s frailty phenotype model includes five components assessed based on objective or subjective report measures [16]. In this study, both methods were used, as recommended by the authors of the Portuguese version of the test [17]. More specifically, physical activity was assessed using a short version of the International Physical Activity Questionnaire [18], developed for older adults (www.ipaq.ki.se). Inactivity and irregular activity were classified as symptoms of frailty. Gait speed was assessed through the 4.6-m walk test, and the best time of the two trials was used for the final score. Symptoms were classified based on cutoff scores of ≥7 and ≥6 s for men and women, respectively.

Weakness was assessed through the handgrip strength test using a dynamometer. The best result of the three trials was used for the final score. Symptoms were classified based on the participants’ gender and body mass index (BMI). The following cutoff scores were used for women: ≤17, ≤17.3, ≤18, and ≤21 for BMI ≤23, 23.1–26, 26.1–29, and >29, respectively. In men, the cutoff scores ≤29, ≤30, and ≤32 were used for BMI ≤24, 24.1–28, and >28, respectively.

Two questions (“I felt that everything I did was an effort” and “I could not get going”) from the Center for Epidemiological Studies-Depression (CES-D) questionnaire [19] were used to assess fatigue. The symptom was classified as present when both statements were evaluated by negative concordance. Weight loss was assessed by subjective report, taking into account the 6-month period prior to assessment. The loss of 4 kg or more was considered an indicator of symptom presence. Frailty status was confirmed by the presence of three to five symptoms and pre-frailty status by the presence of one or two symptoms. In the absence of symptoms, older adults were classified as robust.

Measure of cognitive functioning: Cognitive functioning was assessed using the 6-Item Cognitive Impairment Test (6-CIT) [20]. The 6-CIT is a cognitive screening test composed of six simple questions that assess orientation in time and space, attention and working memory, and verbal memory. The classification of the 6-CIT results as indicative of the presence of changes in cognitive functioning took into account the years of formal education completed by the participants, as proposed by authors of the Portuguese version of the test [21].

### 2.3. Ethical-Legal Considerations

Permission was obtained from the authors of the original version to use the tool and the institutions to conduct the study. The study was approved by the ethics committee of the Health Sciences Research Unit: Nursing, Nursing School of Coimbra, Portugal (decision number P510/06-2018, 510/06-2018). All ethical and legal principles were met. Participation was voluntary, and all participants signed an informed consent form.

### 2.4. Statistical Analyses

Data were analyzed using IBM SPSS Statistics software (version 24, IBM Corp., Armonk, NY, USA). Statistical significance was set at 0.05. Chi-square (χ^2^) tests, Cramer’s V (V) coefficient, Kruskal–Wallis test, Partial eta-squared measure (η^2^_p_), and two-way ANOVA statistics were used. The H-statistic was calculated by summing the squared ranks of a given factor and dividing them by the total mean square for those ranks [22]. Effect size was calculated using η^2^_p_. The Kuder–Richardson Formula 20 (KR-20) was used to assess internal consistency. Spearman’s correlations between SUNFRAIL domain scores and total scores were calculated. Concurrent validity was determined based on Spearman’s correlations between SUNFRAIL total score and the number of chronic conditions and medication intake. The Receiver Operating Characteristic (ROC) curve was plotted to compare the sensitivity and specificity of different cutoff point(s) for frailty screening. The SUNFRAIL score was used as a test variable and the absence/presence of the Fried frailty criteria as a state variable. The Youden index was calculated to select the optimal cutoff point. The area under the curve (AUC) with 95% confidence interval and other summary measures of test accuracy were also reported.

## 3. Results

### 3.1. Sociodemographic and Clinical Characteristics

The participants were mostly female (*n* = 98), with a mean age of 71.09 ± 7.85 years and a mean education level of 8.08 ± 4.17 years (Table 1). According to Fried’s diagnostic criteria for frailty, of the 128 older adults, 23 were frail, 53 were pre-frail, and 52 were robust. Table 2 shows that reduced strength was the most common symptom among frail and pre-frail participants. A large percentage of frail older adults showed reduced speed, activity, and fatigue; however, only one frail person reported unintentional weight loss. Interestingly, BMI in this group was quite high, reaching a mean value of 29.62 (±6.44). For pre-frail participants, reduced activity and reduced speed were the second and third most common symptoms. Fatigue was reported by 8% of pre-frail older adults, and none of them confirmed unintentional weight loss (Table 2). In this group, BMI reached a mean value of 26.96 (±4.02), which was similar to the mean value found in the group of robust older adults (26.59 ± 3.93). Multimorbidity (presence of two or more chronic conditions) was reported in 90% of participants. Frail participants had, on average, more chronic conditions than robust or pre-frail participants (Table 1). Eighty-four percent of participants reported taking medication, and more than half of the sample (53%) reported being polymedicated (two or more drugs) (Table 1).

### 3.2. Internal Consistency of the SUNFRAIL Tool

Internal consistency was low (0.522). The correlations between the SUNFRAIL total score and the three domain scores were statistically significant (*p* < 0.001). The correlation between the total score and the biological domain score was strong (rho = 0.84), and both the correlations between the total score and the psychological domain score (rho = 0.65) and the total score and the social domain score (rho = 0.55) were moderate. The correlations between the SUNFRAIL domains were significant but weak. The highest correlation was found between the biological and the social domain scores (rho = 0.29; *p* = 0.001), and the lowest between the psychological and the social domain scores (rho = 0.22; *p* = 0.011). The rho coefficient for the biological and the psychological domain scores was 0.25 (*p* = 0.004) (Table 2).

### 3.3. SUNFRAIL Score in Robust and Non-Robust Older Adults

The analysis of the SUNFRAIL total score using the Kruskal–Wallis test revealed statistically significant differences between groups (H(2) = 21.708; *p* < 0.001). The multiple comparisons of mean ranks showed that robust participants scored significantly lower on the SUNFRAIL tool than pre-frail (*p* = 0.048) and frail (*p* < 0.001) participants. Significant differences were also found between the scores obtained by frail and pre-frail older adults (*p* = 0.001). The effect size was medium (η^2^_p_ = 0.171). Significant between-group differences were found in the biological (H(2) = 22.385; *p* < 0.001; η^2^_p_ = 0.176) and the psychological (H(2) = 10.743; *p* = 0.005; η^2^_p_ = 0.085) domains of the SUNFRAIL tool, but not in the social domain (H(2) = 4.860; *p* = 0.088).

Moreover, the multiple comparisons of mean ranks showed significant differences between robust and frail older adults in both biological (*p* < 0.001) and psychological (*p* = 0.008) domains. Significant differences between robust and pre-frail participants were found in the psychological domain (*p* = 0.038) but not in the biological domain. Significant differences between pre-frail and frail older adults were only found in the biological domain (*p* < 0.001) (Figure 1).

### 3.4. SUNFRAIL Tool and 6-CIT

Seventy-three percent of participants showed no significant cognitive changes. Older adults with and without cognitive decline were not equally distributed in the robust, pre-frail, and frail groups (χ^2^ = 12.932; *p* = 0.002; Vc = 0.3) (Table 1). A non-parametric two-way ANOVA was used to analyze the effect of cognitive status on SUNFRAIL scores. The correlation between frailty status (robust, pre-frail, and frail) and cognitive status (without cognitive decline and with mild cognitive decline) was statistically significant (H(2) = 22.138, *p* < 0.001, η^2^_p_ = 0.174), explaining 17.4% of total variance. In terms of main effect, the cognitive status did not contribute to the distribution of the SUNFRAIL score (H(1) = 0.431, *p* = 0.51, η^2^_p_ = 0.004). Frailty status proved to have a significant effect on the distribution of the SUNFRAIL score (H(1) = 18.095, *p* < 0.001, η^2^_p_ = 0.147), explaining 14.7% of total variance.

### 3.5. Concurrent Validity of the SUNFRAIL Tool

Frail participants had, on average, more chronic conditions than robust or pre-frail participants, but these differences (Kruskal–Wallis test) were not significant (*p* > 0.05). The number of chronic conditions correlated significantly but moderately with the total score of the SUNFRAIL tool (rho = 0.44; *p* = 0.01).

### 3.6. Sensitivity and Specificity of the SUNFRAIL Tool

Figure 2 shows the ROC curve for the SUNFRAIL score, using Fried’s frailty criteria (absence of symptoms vs. presence of symptoms) as the gold standard.

The AUC was 0.671 (95% CI = 0.58–0.77; *p* < 0.01). The cutoff point >2 had the best sensitivity and specificity (Table 3). Predictive values and likelihood ratios for this cutoff point are shown in Table 4.

## 4. Discussion

The low internal consistency of the Portuguese European version of the SUNFRAIL tool may have been due to the reduced number of items representing three different domains of individual functioning. On the other hand, significant moderate (psychological and social) to strong (biological) correlations were found between domain scores and total score, proving that the instrument may collect relevant data for defining the follow-up care plan. The correlations between different SUNFRAIL domains showed more satisfactory results than those found in the study conducted in the Netherlands [13]. In the latter study, no significant correlations were found between the physical and the social domains in the SUNFRAIL tool; the internal consistency value was also not reported.

The results on reliability require further discussion on the SUNFRAIL structure. Although it is an easy-to-use instrument covering three different domains, it may be necessary to increase the number of items in the psychological and social domains. Concerning the social domain, it is important to address older adults’ perceived satisfaction with the available social support rather than only questioning if such support exists or not.

The findings on internal consistency also suggest the need for reviewing item content and scoring options. During the construct validity process, lay, research, and clinical practice communities raised questions about the difficulty in understanding some items. Based on their feedback, these items were reformulated [15]. Still, some of them may be ambiguous for the Portuguese context, probably undermining the tool’s consistency. In our opinion, items 2, 4, and 9 are more sensitive to cultural issues and items 1 and 7 are more sensitive to the meaning attributed to them. Therefore, we recommend further research on older adults’ understanding of the items and how their answers can be influenced by social desirability. As for item scoring, the dichotomous (yes/no) response option makes it easier to use. However, multiple-choice questions may enable a more reliable screening. The training of health and social care professionals in the SUNFRAIL administration may also be helpful. The instrument could also include a detailed description of each item to check if the meaning attributed by the interviewees to these items is the same as that intended by the tool authors. It could also provide guidance on the care-pathways to be suggested or activated in response to the symptoms and information on the available resources, as suggested by other studies [8].

As some SUNFRAIL items seem to be culturally sensitive, the proposed approach may benefit older people by raising their awareness about health changes, which, despite being warning signs, are often assumed as “normal”, preventing the search for timely help. The fact that there is an over expression of obesity over weight loss deserved our attention. Several studies [23,24] show that weight loss but also obesity can be a frailty indicator. This aspect may be dependent on the cultural context and deserves further reflection. Although the SUNFRAIL tool can still be improved, we believe it may allow in-depth data collection. One of the strengths of the European Portuguese version [15] is its ability to discriminate between robust, pre-frail, and frail older adults. These results are in line with another study [13]. The differences between groups were predominantly higher in the biological domain and lower in the other domains, which may reinforce the idea that the biological domain is overrated in comparison with the psychological and the social domains, which is also similar to the results found by Gobbens et al. (2012) [7].

The total explained variance of the SUNFRAIL was higher for the interaction between frailty status and cognitive status than when the cognitive status or the frailty status were analyzed per se, which, in our opinion, may reinforce the multidimensionality of frailty [13]. As regards concurrent validity, the SUNFRAIL was significantly correlated with multimorbidity. Previous studies [5,6,7] also suggested a high prevalence of multiple chronic diseases in frail older adults. Other authors [25] also argue that a mean number of chronic diseases is a relevant determinant of frailty, with overlap rates of frailty and multimorbidity reaching 25%. We are able to determine two-thirds for the cutoff point, with score 2 indicating a robust health status and score 3 or more indicating pre-frailty or frailty, which is consistent with a previous study [13]. The sensitivity and specificity values for a pilot study are satisfactory, but more studies are needed. For being a self-assessment questionnaire, some domains can be underestimated. Carers or other significant people should be included for a more comprehensive assessment.

### Strengths and Limitation of the Study

This study’s major strength is that it determined the validity and reliability of this European Portuguese version of the SUNFRAIL tool. This cross-cultural adaptation process followed rigorous quality procedures, resulting in the first version of an instrument for frailty screening suitable for community-dwelling older people and can be easily used in primary health care settings. A most important limitation of this study is the sample size, especially the reduced number of participants with frailty. The latter is due to the fact that most study participants were involved in physical and social activities in the community, which makes them less frail. This fact may have conditioned the instrument’s results on sensitivity and specificity. As so, future research should focus both on older adults who maintain physical and cognitive activity and on older adults who are no longer active. Future research should also examine the SUNFRAIL performance in groups of older adults who are socially involved with those who are socially isolated, to obtain more accurate data on the social and psychological domains of the tool. The analysis of SUNFRAIL that considers the distribution of participants into different age groups is additionally recommended. This analysis would enable to verify whether the cutoff score suggested in the present study also applies to the oldest older adults. Finally, other studies should be conducted to reinforce the validity (construct validity against other frailty tests validated for the Portuguese population, concurrent and predictive validity) and reliability (temporal stability) of the European Portuguese version of the SUNFRAIL tool, as well as to explore its association with adverse health outcomes.

## 5. Conclusions

The European Portuguese version of the SUNFRAIL tool is a promising frailty screening tool for clinical practice. It is an easy-to-use and friendly instrument. More consistent results on validity and reliability are needed for its use in clinical practice nationwide.

## Figures and Tables

**Figure 1 ijerph-18-01394-f001:**
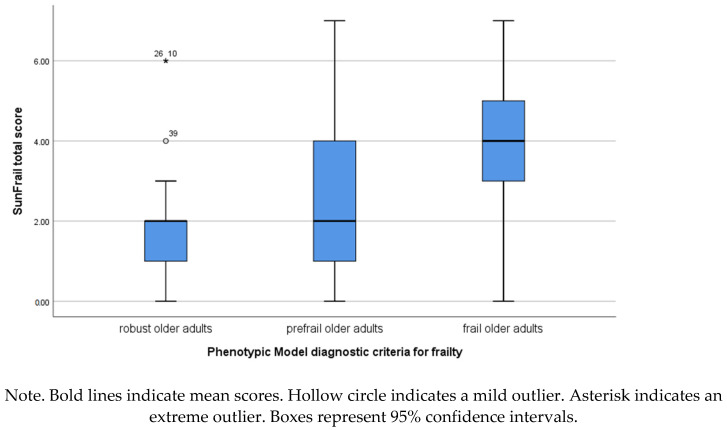
SUNFRAIL score for participants classified as robust, pre-frail, and frail based on Fried’s Phenotype Model.

**Figure 2 ijerph-18-01394-f002:**
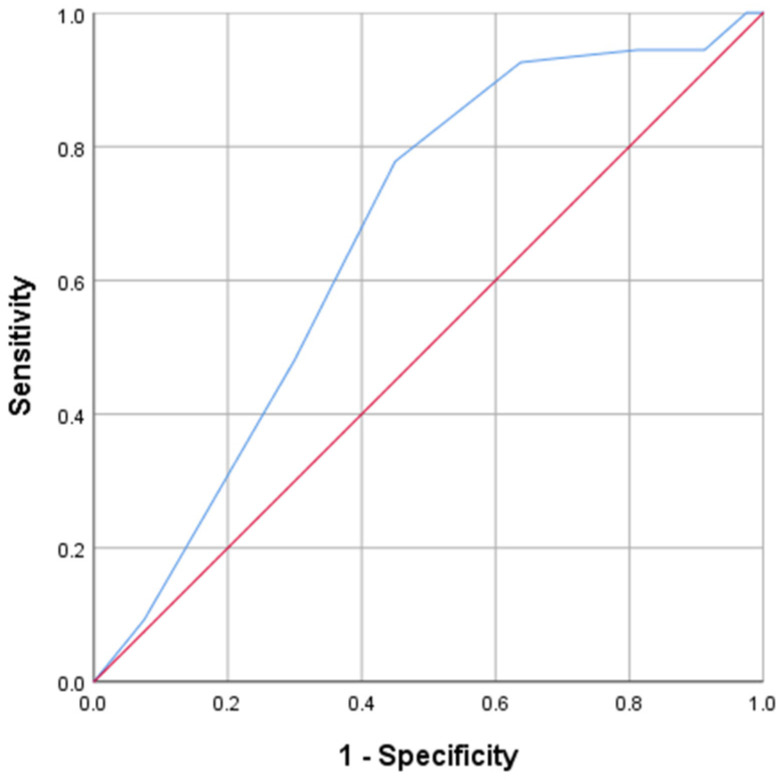
Receiver Operating Characteristic (ROC) curve for the SUNFRAIL tool, using diagnostic criteria for frailty from the Fried phenotype model as a gold standard.

**Table 1 ijerph-18-01394-t001:** Demographic, clinical, and neuropsychological characteristics of the sample.

	Robust Older Adults (*n* = 54)	Pre-Frail Older Adults (*n* = 56)	Frail Older Adults (*n* = 24)		
	%	%	%	χ^2^	Cramer’s *V*
Gender: Female/Male	72/28	80/20	75/25	1.021	0.600
Marital Status: single/married/widowed/divorced	4/69/13.5 /13.5	4/54.5/36 /5.5	4/46/37.5 /12.5	9.628	0.141
6-CIT: without cognitive decline/with mild cognitive decline	81.5/18.5	79/21	42/58	14.853	0.001
	Mean (SD)	Mean (SD)	Mean (SD)	Kruskal-Wallis (*p*)	Pairwise comparisons
Age	70.83 (4.50)	72.15 (5.36)	72.39 (3.86)	0.321	--------
Education level	8.76 (3.99)	7.84 (4.14)	7.25 (4.20)	0.299	--------
Medication intake	1.5 (1.28)	1.91 (1.37)	2.63 (1.31)	0.003	R < F *
Comorbidities	3.81 (1.84)	3.84 (2.09)	4.88 (1.87)	0.078	--------

F: frail older adults; R: robust older adults; * *p* < 0.01; η^2^_p_ = 0.089.

**Table 2 ijerph-18-01394-t002:** SUNFRAIL score and diagnostic criteria for frailty based on Fried’s Phenotype Model.

			Non-Robust Older Adults		
		Robust Older Adults (*n* = 54)	Total (*n* = 80)	Pre-Frail Older Adults (*n* = 56)	Frail Older Adults (*n* = 24)
SUNFRAIL Total score	Mean ± SD (range)	1.83 ± 1.37 (0–6)	2.84 ± 1.81 (0–7)	2.36 ± 1.63 (0–7)	3.96 ± 1.73 (0–7)
SUNFRAIL -Biological	Mean ± SD (range)	1.07 ± 0.91 (0–4)	1.56 ± 1.21 (0–4)	1.18 ± 1.05 (0–4)	2.46 ± 1.10 (0–4)
SUNFRAIL -Psychological	Mean ± SD (range)	0.57 ± 0.69 (0–2)	0.94 ± 0.68 (0–2)	0.88 ± 0.69 (0–2)	1.08 ± 0.65 (0–2)
SUNFRAIL -Social	Mean ± SD (range)	0.19 ± 0.48 (0–2)	0.34 ± 0.50 (0–2)	0.30 ± 0.50 (0–2)	0.42 ± 0.50 (0–1)
Fried’s Phenotype Model criteria	Mean ± SD (range)	0.00 ± 0.00 (0.00–0.00)	1.89 ± 1.06 (1–4)	1.27 ± 0.45 (1–2)	3.33 ± 0.48 (3–4)
Weight loss	% of persons with symptom	-------	1.32%	0.00%	4.35%
Fatigue	% of persons with symptom	-------	27.63%	7.55%	73.91%
Reduced activity	% of persons with symptom	-------	51.32%	39.62%	78.26%
Reduced speed	% of persons with symptom	-------	40.79%	20.75%	86.96%
Reduced hangrip strength	% of persons with symptom	-------	75.00%	66.04%	95.65%

**Table 3 ijerph-18-01394-t003:** Sensitivity, specificity, and Youden Index of the SUNFRAIL tool.

Cutoff Point	Sensitivity	Specificity	Youden Index
>0	9.26%	92.50%	0.02
>1	48.15%	70.00%	0.18
**>2**	**77.78%**	**55.00%**	**0.33**
>3	92.59%	36.25%	0.29
>4	94.44%	18.75%	0.13
>5	94.44%	8.75%	0.03
>6	100.00%	2.50%	0.02
>8	100.00%	0.00%	0.00

Note. In bold: cutoff points, sensitivity, and specificity for the maximal Youden Index.

**Table 4 ijerph-18-01394-t004:** Screening properties of the SUNFRAIL tool for cutoff >2.

Sensitivity	Specificity	AUC	PPV	NPV	LR+	LR-
0.78 (0.64–0.88)	0.55 (0.43–0.66)	0.666 * (0.57–0.76)	0.79 (0.68–0.86)	0.54 (0.47–0.61)	2.48 (1.45–4.23)	0.58 (0.44–0.77)

Note. AUC: area under the curve; LR-: negative likelihood ratio; LR+: positive likelihood ratio; NPV: negative predictive value; PPV: positive predictive value. Numbers in parentheses show a 95% confidence interval. * *p* < 0.01.

## Data Availability

The data presented in this study are available on request from the corresponding author. The data are not publicly available because this issue was not considered within the informed consent signed by the participants of the study.

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
