# Peer review of "Validation and Screening Capacity of the European Portuguese Version of the SUNFRAIL Tool for Community-Dwelling Older Adults"

_ijerph, 2021, doi:10.3390/ijerph18041394_

Round 1
Reviewer 1 Report
In this paper, authors present a cross-sectional pilot study in a convenience sample of 128 community-dwelling older adults. Internal consistency and concurrent validity were examined, as well as sensitivity and specificity (ROC curve analysis). The reported results suggest The European Portuguese version of the SUNFRAIL tool is a promising frailty screening tool for community-dwelling older adults to be routinely used in clinical practice. However, more consistent results on its validity and reliability are needed to be used nationwide.
However, the reviewer does not understand, what is the goal of the study, since similar studies have been done before. Moreover, the reviewer suggests some changes, please check it as following.
I wonder if the title of the manuscript is appropriate because the validation of the Sunfrail questionnaire into Portuguese was done in an earlier work. This work was to compare the results of two separate Frailty rating scales (Sunfrali & Fried), so does the title of the work reflect what the authors actually examined?
I propose to rewrite the manuscript by supplementing it with the results of Fried'a questionnaire, providing the results of the IPAQ questionnaire and the walk test, and on this basis to formulate new conclusions.
Because the conclusions contained in this manuscript are consistent with the previous works of the authors.
Line 3 – I propose – Community
- Materials and Methods
Please explain why the six-minute walk test was used?
By what methods were the cognitive functions assessed, Table 1 shows that 6-CIT was used
Which version of the IPAQ was used in the study?
line 93 - I propose to add a Statistical analysis paragraph
- Results
Sorry, but I don't understand Table 2, I can only see the results of the SUNFRAIL questionnaire, I miss the results of the Fried questionnaire, please explain this
I do not see data on walking parameters, physical activity and grip strength, have they been analyzed?
Please improve the quality of Figure 1 as it is illegible
- Discussion
line 241-244 - please support this statement with data from the references
line 247 - 249 - Please cite previous studies supporting this statement.
Author Response
Reviewer 1:
- However, the reviewer does not understand, what is the goal of the study, since similar studies have been done before. Moreover, the reviewer suggests some changes, please check it as following. I wonder if the title of the manuscript is appropriate because the validation of the Sunfrail questionnaire into Portuguese was done in an earlier work. This work was to compare the results of two separate Frailty rating scales (Sunfrali & Fried), so does the title of the work reflect what the authors actually examined? To our knowledge, there are no validation studies on SUNFRAIL for Portuguese population. The only publication we know is related to the process of cross- cultural adaptation of the tool and is of our authorship. The present study is a continuation of the referred study. If there is another European Portuguese version of the SUNFRAIL, we would like to know it. Thank you. In any case, the objectives and title have been reformulated, as suggested, to facilitate reading.
- I propose to rewrite the manuscript by supplementing it with the results of Fried'a questionnaire, providing the results of the IPAQ questionnaire and the walk test, and on this basis to formulate new conclusions.
- Following the suggestion, more detailed information about the performance on this test was added (section Results and table 2). Taking into account that the focus of our study was to define the screening capacity of the SUNFRAIL against the FRIED’s phenotypic criteria, we consider that the information provided is sufficient.
- Please explain why the six-minute walk test was used? The information about the six-minute test was incorrect. The assessment of walking speed was based on the 4.6-meter test. The information was corrected and the details about cut-off points for men and women were added.
- By what methods were the cognitive functions assessed, Table 1 shows that 6-CIT was used. We rewrote the paragraph presenting 6-CIT (Method section) to make this information clearer.
- Which version of the IPAQ was used in the study? Following the suggestion, more detailed information about this component was provided. We have also included more details on other four components of the Fried test. These details are related to the administration procedures as well as the procedures to decide whether the symptom is present or absent.
- Sorry, but I don't understand Table 2, I can only see the results of the SUNFRAIL questionnaire, I miss the results of the Fried questionnaire, please explain this. In the original version of the manuscript, data related to the Fried test was presented in the line entitled “Phenotypic Model criteria”. To facilitate reading, the name of line was changed for “Fried´s Phenotypic Model criteria”.
- I do not see data on walking parameters, physical activity and grip strength, have they been analyzed? Following the suggestion, we include data on walking parameters, physical activity and grip strength on narrative (section Results) and table 2.
- Please improve the quality of Figure 1 as it is illegible. We replace the Figure 1 to improve its eligibility.
- line 241-244 - please support this statement with data from the references. The required information was added.
- line 247 - 249 - Please cite previous studies supporting this statement. We added the references, as suggested.
Reviewer 2 Report
Overall an interesting manuscript. Authors used a scientific approach suitable for the study. Furthermore, they clearly reported that more consistent results on validity and reliability are needed for clinicians to use the Portuguese version of the SUNFRAIL.
I have only some comments that I report below.
Introduction
Line 35-36: I suggest considering a paper that analyzed changes in physical activity in active older people during the COVID-19 pandemic. Please consider the following article: “Giustino V. et al. Physical Activity Levels and Related Energy Expenditure during COVID-19 Quarantine among the Sicilian Active Population: A Cross-Sectional Online Survey Study. Sustainability 2020, 12, 4356.”
Line 36-38: As above, please provide a reference for this statement. I suggest considering the following article: “Suzuki Y. et al. Physical Activity Changes and Its Risk Factors among Community-Dwelling Japanese Older Adults during the COVID-19 Epidemic: Associations with Subjective Well-Being and Health-Related Quality of Life. Int J Environ Res Public Health 2020, 10, 17(18):6591.”
Materials and Methods
Authors should better present the collected data. In particular, after reporting the type of study and the exclusion criteria, I suggest reporting the inclusion criteria, the sociodemographic and clinical data collected (lines 77-86), and then the tests performed, and the questionnaires administered.
How did Reserchers proceed for the translation of the SUNFRAIL tool?
Please, provide details of the handgrip test performed (procedure and instrument used).
Please provide information on the questionnaire used to assess fatigue.
Authors stated: “Weight loss was assessed by subjective report.” (line 72). What was the reported weight loss period?
Discussion
Authors should report limitations and strengths of the study in a dedicated paragraph.
Author Response
Line 35-36: I suggest considering a paper that analyzed changes in physical activity in active older people during the COVID-19 pandemic. Please consider the following article: “Giustino V. et al. Physical Activity Levels and Related Energy Expenditure during COVID-19 Quarantine among the Sicilian Active Population: A Cross-Sectional Online Survey Study. Sustainability 2020, 12, 4356.” Line 36-38: As above, please provide a reference for this statement. I suggest considering the following article: “Suzuki Y. et al. Physical Activity Changes and Its Risk Factors among Community-Dwelling Japanese Older Adults during the COVID-19 Epidemic: Associations with Subjective Well-Being and Health-Related Quality of Life. Int J Environ Res Public Health 2020, 10, 17(18):6591.”. Following the suggestion, we considered the studies cited in our paper, adding the information of interest in the background section.
- Authors should better present the collected data. In particular, after reporting the type of study and the exclusion criteria, I suggest reporting the inclusion criteria, the sociodemographic and clinical data collected (lines 77-86), and then the tests performed, and the questionnaires administered. Following the suggestion, we reorganized the method section and included subheadings to make reading easier.
- How did Researchers proceed for the translation of the SUNFRAIL tool? The information about the translation and adaptation processes of the SUNFRAIL tool are published elsewhere. We rewrote the last paragraph of the introduction section to make this information clearer.
- Please, provide details of the handgrip test performed (procedure and instrument used). Following the suggestion, more detailed information about this component was provided. We have also included more details on other four components of the Fried test. These details are related to the administration procedures as well as the procedures to decide whether the symptom is present or absent.
- Please provide information on the questionnaire used to assess fatigue. Following the suggestion, more detailed information about this component was provided. We have also included more details on other four components of the Fried test. These details are related to the administration procedures as well as the procedures to decide whether the symptom is present or absent.
- Authors stated: “Weight loss was assessed by subjective report.” (line 72). What was the reported weight loss period? Following the suggestion, more detailed information about this component was provided. We have also included more details on other four components of the Fried test. These details are related to the administration procedures as well as the procedures to decide whether the symptom is present or absent.
- Authors should report limitations and strengths of the study in a dedicated paragraph. The required information was added.

Round 2
Reviewer 1 Report
Thank you to the authors, I am satisfied
Author Response
Dear Reviewer,
Thank you so much for your feedback on the changes that we made,